# Neurocognitive Performance Improvement after Obstructive Sleep Apnea Treatment: State of the Art

**DOI:** 10.3390/bs11120180

**Published:** 2021-12-16

**Authors:** Isabella Pollicina, Antonino Maniaci, Jerome R. Lechien, Giannicola Iannella, Claudio Vicini, Giovanni Cammaroto, Angelo Cannavicci, Giuseppe Magliulo, Annalisa Pace, Salvatore Cocuzza, Milena Di Luca, Giovanna Stilo, Paola Di Mauro, Maria Rita Bianco, Paolo Murabito, Vittoria Bannò, Ignazio La Mantia

**Affiliations:** 1Department of Medical and Surgical Sciences and Advanced Technologies “GF Ingrassia”, ENT Section, University of Catania, 95123 Catania, Italy; isabellapollicina90@gmail.com (I.P.); antonino.maniaciphd.unict.it (A.M); s.cocuzza@unict.it (S.C.); milenadilluca88@gmail.com (M.D.L.); giovastilo@gmail.com (G.S.); paola_mp86@hotmail.it (P.D.M.); vittoriabanno@gmail.com (V.B.); igolama@gmail.com (I.L.M.); 2Department of Human Anatomy and Experimental Oncology, Faculty of Medicine, UMONS Research Institute for Health Sciences and Technology, University of Mons (UMons), 7000 Mons, Belgium; antonino.maniaci@phd.unict.it (A.M.); jerome.lechien@umons.ac.be (J.R.L.); 3Department of Otorhinolaryngology and Head and Neck Surgery, Foch Hospital, School of Medicine, UFRSimone Veil, Université Versailles Saint-Quentin-en-Yvelines (Paris Saclay University), 91190 Paris, France; 4Department of Otorhinolaryngology and Head and Neck Surgery, CHU de Bruxelles, CHU Saint-Pierre, School of Medicine, Université Libre de Bruxelles, 1050 Brussels, Belgium; 5Oral Surgery Unit, Department of Head-Neck Surgery, Otolaryngology, Head-Neck, Morgagni Pierantoni Hospital, 47121 Forli, Italy; claudio@claudiovicini.com (C.V.); giovanni.cammaroto@hotmail.com (G.C.); angelo.cannavicci@auslromagna.it (A.C.); giuseppe.magliulo@uniroma1.it (G.M.); 6Department of “Organi di Senso”, University “Sapienza”, 00185 Rome, Italy; annalisapace90@gmail.com; 7Otolaryngology, Department of Health Science, University Magna Graecia of Catanzaro, Viale Europa, Germaneto, 88100 Catanzaro, Italy; biancomariarita@hotmail.it; 8Department of Surgery and Medical and Surgical Specialties—Section of Anaesthesiology and Intensive Care, University of Catania (Italy), 95123 Catania, Italy; paolomurabito@tiscali.it

**Keywords:** OSAS, cognitive impairment, CPAP, MAD, positional OSA

## Abstract

Background: Obstructive Sleep Apnea (OSA) syndrome is a respiratory sleep disorder characterized by partial or complete episodes of upper airway collapse with reduction or complete cessation of airflow. Although the connection remains debated, several mechanisms such as intermittent hypoxemia, sleep deprivation, hypercapnia disruption of the hypothalamic–pituitary–adrenal axis have been associated with poor neurocognitive performance. Different treatments have been proposed to treat OSAS patients as continuous positive airway pressure (CPAP), mandibular advancement devices (MAD), surgery; however, the effect on neurocognitive functions is still debated. This article presents the effect of OSAS treatments on neurocognitive performance by reviewing the literature. Methods: We performed a comprehensive review of the English language over the past 20 years using the following keywords: neurocognitive performance and sleep apnea, neurocognitive improvement and CPAP, OSAS, and cognitive dysfunction. We included in the analysis papers that correlated OSA treatment with neurocognitive performance improvement. All validated tests used to measure different neurocognitive performance improvements were considered. Results: Seventy papers reported neurocognitive Performance improvement in OSA patients after CPAP therapy. Eighty percent of studies found improved executive functions such as verbal fluency or working memory, with partial neural recovery at long-term follow-up. One article compared the effect of MAD, CPAP treatment on cognitive disorders, reporting better improvement of CPAP and MAD than placebo in cognitive function. Conclusions: CPAP treatment seems to improve cognitive defects associated with OSA. Limited studies have evaluated the effects of the other therapies on cognitive function.

## 1. Introduction

Obstructive Sleep Apnea (OSA) syndrome is a respiratory sleep disorder characterized by partial or complete recurrent episodes of upper airway collapse during the night. OSA is a frequent and often underestimated pathology affecting between 2% and 5% of middle-aged population OSAS with excessive daytime sleepiness occurred in 6% (range, 3–18%) of men and 4% (range, 1–17%) of women [1,2,3]. The prevalence increased with time, and OSAS was reported in 37% of men and 50% of women [3].

Sleep obstructive breathing disorders represent a topic that has aroused interest for more than two decades, whose treatment was initially possible only to those who had purchasing power; to date, however, considering the very high prevalence in the general population, the first approach proposed is prevention through the association with adequate lifestyles.

As early as 2001, Duran et al. had estimated the prevalence and related clinical characteristics in the general population through a cross-sectional study that included 2148 subjects in the first phase. While the authors found habitual snoring found in 35% of the population, apneas reached 6% [4].

OSA is a growing health problem increasingly affecting the adult population, which is associated with several metabolic and cardiovascular injuries, neurocognitive impairment, and behavioral disorders brain functional [5,6,7,8]. Consistently, OSA alters cognitive performances (such as memory and attention) by reducing sleep quality and continuity [1]. Moreover, chronic sleep deprivation associated with OSA may promote toxic products that induce neurodegeneration (Figure 1) [9].

A long-term effect of OSAS is a decrease in gray matter in the hippocampus, anterior cingulate cortex, ventrolateral frontal cortex, and a portion of the cerebellar cortex [10,11,12]. Devita et al. reported morphological changes, such as brain tissue damage, associated with impairment of cognitive functions, including attention, executive functioning, motor efficiency, working memory, and long-term episodic memory, in patients with OSAS [5].

Currently, different treatment approaches have been proposed to treat OSAS patients as continuous positive airway pressure (CPAP), mandibular advancement devices (MAD), positional, or surgical treatment with efficacy still debated [10,11,13,14,15,16]. Rosenzweig et al., comparing cognitive functions and neurostructural changes after 1 month of treatment in 68 OSA patients, 34 of which were treated with CPAP with BSC (best supportive care), while 34 were only CPAP treated, demonstrated a potential strengthening in interregional connectivity between the non-dominant thalamus with the hippocampus and cerebellar cortex only in CPAP treated group [17]. In addition, in the same group of patients, improvement in sleepiness (showed by the ESS) was strongly positively correlated to changes occurring in the brainstem during this period.

Instead, the results obtained by Lim et al. reported no significant difference in the overall deficits between 2 weeks of CPAP treatment, overnight oxygen supplementation, and placebo-CPAP [18]. However, the authors hypothesized the beneficial effect of CPAP on information processing speed, alertness, and sustained attention.

This article presents the effect of CPAP treatment in OSA patients on neurocognitive performance improvement by a comprehensive review of the literature.

## 2. Materials and Methods

### 2.1. Protocol Data Extraction and Outcomes

The authors I.P. and A.M. analyzed the data of the literature. The study team members solved any disagreements via a discussion. Thus, the studies included were analyzed to obtain all the available data and guarantee eligibility among subjects enrolled. The authors collected ‘patients’ features such as symptoms, age, and neurocognitive function assessed through validated questionnaires and treatment modalities. Moreover, we also collected: author data, year, study design, sample size, statistical analysis, main findings, and conclusions. If the required data were not complete, we contacted the authors of the included studies through correspondence by the author’s email or Research Gate. All studies that met the following criteria were included:Original articles;The article was published in the English language;The studies included patients with impaired neurocognitive performance undergoing treatment for OSAS;The studies reported detailed information on pre-and post-treatment OSAS outcomes, validated questionnaires on neurocognitive performance at baseline and after treatment, and patients’ comorbidities;We excluded case reports, editorials, letters to the editor, or reviews from the study.

### 2.2. Electronic Database Search

We searched on electronic PubMed, Scopus, and Web of Science databases for studies on patients with OSAS and impaired neurocognitive function undergoing treatment for sleep apnea of the last 20 years of literature (from 1 July 2001 to 1 July 2021) by two different authors (I.P. and A.M.), through MeSH, related keywords, and Entry Terms. The related search keywords used were: “Neurocognitive function,” obstructive sleep apnea treatment, “cognitive function and sleep apnea”, “obstructive sleep apnea”, “cognitive function and oral appliance”, “Mandibular advancement and cognitive performance”, “Neurocognitive function and obstructive sleep apnea”, “Sleep-disordered breathing and neurocognitive function”, “neurocognitive improvement and CPAP”, “OSAS and cognitive dysfunction”. The PICOTS statements for the method presentation [19]. We considered Participants (OSAS patients); Intervention (CPAP and oral appliance); Control (OSAS patients not treated); Outcome (improved neurocognitive performance after therapy); and study type (observational study). We imposed language, publication date, and publication status as restrictions. We considered the primary outcome a significant improvement in the subjective questionnaires administered to the patient post-treatment follow-up. Other parameters assessed in the studies were considered secondary outcomes. We also considered the “Related articles” option on the PubMed homepage. We used Reference manager software (EndNote X7^®^, Thomson Reuters, Philadelphia, PA, USA) to collect references and remove duplicates. Thus, the investigators examined titles and abstracts of papers available in the English language. The full texts identified were thus screened for original data. We retrieved the related references while other relevant studies were checked manually.

## 3. Results

We retrieved and analyzed 17 articles, of which 14 papers assessed neurocognitive performance in OSA patients before and after CPAP treatment, while 3 papers assessed the degree of cognitive impairment in untreated OSA patients (Figure 2).

Among studies retrieved, 80% reported improved executive functions such as verbal fluency or working memory, with partial neural recovery at long-term follow-up. In all studies, surgery and CPAP were considered effective in executive function and attention, as summarized in Table 1 and Table 2.

### 3.1. Attention and Executive Functions

Executive dysfunction could be explained by the capacity of attention deficit, slow processing speed, and impaired short-term memory, perhaps influenced by drowsiness. Thus, executive functions may be impaired in patients with OSA patients not adequately treated. Nine papers reported greater improvement of attention and vigilance after CPAP treatment [1,8,17,20,21,22,23,24]. Lau et al. compared a group of 37 OSA patients with respiratory disturbance index (RDI) >15 treated with CPAP to a control group of healthy patients, which showed the OSAgroup’s worse performance on neuropsychological measures of complex attention and executive functions evaluated through Stroop interference score and Wisconsin Cord Sorting Test [8]. Additionally, Castronovo et al., in a controlled prospective study, reported how the OSA group after CPAP treatment compared to the healthy control group reported worse results on data referring to neurocognitive assessing attention and executive functions after treatment (Stroop test: 5.08 ± 3.32 pre-treatment, 0.83 ± 1.53 after treatment, *p* < 0.001; Trial Making test B: 82.15 ± 26.16 pre-treatment, 78.85 ± 23.42 post-treatment, *p* < 0.001) [20]. The OSA patients included in the study had no comorbidities and AHI ≥ 30. Turner et al. found no significant change in Attentional Matrices (*p* = 0.55) comparing test results before and after CPAP treatment in 16 patients with mean AHI of 30.92 ± 11.21 [11].

### 3.2. Memory and Learning

Learning and memory are closely related and can be divided into verbal, visual, and procedural domains, and there are several methods to evaluate this function [10,22]. Turner et al. found statistically significant improvements in cognitive functioning, such as working memory (Digit Span Backward, *p* = 0.004), long-term verbal memory (Short Story Test, *p* = 0.004), and short-term visuospatial memory (Corsi Span, *p* = 0.02) after 3 months of CPAP therapy [11]. They observed no significant changes in the Digit Span Forward (*p* = 0.61) and ROCF/Modified Taylor Complex Figure (copy: *p* = 0.80; recall: *p* = 0.48) performances. Ng et al., in a prospective controlled study, compared test results in 30 OSA patients treated with home CPAP before and after 12 months of treatment and found a significant improvement of Digit symbol score after treatment (baseline: 30.4 ± 12.2, post 12 months CPAP: 35.7 ± 15.1). The mean AHI of the OSA patient user CAP was 16.8 ± 14.2 [22]. Lim W et al. demonstrated that only two weeks of CPAP therapy was insufficient to show a significant effect on cognitive function [16]. Among the tests battery submitted to patients at baseline and after treatment with CPAP, only Digit Vigilance Time showed significant improvement specific to CPAP treatment (baseline: 5.6, post CPAP: 7.2, *p* = 0.020).

### 3.3. General Cognitive Function

Kanbay, A. et al. evaluated cognitive functions in 33 OSA patients compared to 17 controls using a mini-mental state examination test (MMSE) [21]. Baseline MMSE evaluation scores were lower in OSAS patients than in the control group (23.5 ± 3.6 and 28.1 ± 1.4, respectively; *p* = 0.0001). A significant improvement in MMSE scores was detected 3 months after starting CPAP (26.5 ± 2.8, *p* = 0.001). Liguori et al. reported a higher score at the Raven test in patients treated with CPAP with an average AHI after treatment 3.14 ± 1.54 compared to the OSAS group not treated with a mean AHI 36.34 ± 11.42 [1]. Raven’s Advanced Progressive Matrices is a neurocognitive test that evaluates non-verbal intelligence, visual processing speed, cognitive speed, and flexibility [1]. Canessa et al. observed a significant improvement in cognitive domains tested by Raven in OSA patients after 3 months of treatment with CPAP, correlated to a significant reduction in AHI (before treatment AHI mean: 55.83 ± 19.08; after 3 months CPAP AHI mean: 2.5 ± 2.4 [9].

### 3.4. Psychomotor Function

The psychomotor function represents neurocognitive processing speed and is often measured by two-hand coordination or reaction times. Kalcina et al. reported an impaired speed of perception, convergent, and operative thinking in untreated OSA patients compared to the control group measured by CRD-series test (CRD5.2 ± 1.8 vs. 4.5 ± 1.33, *p* < 0.01; CRD. 41: OSA 25.1 ± 17.99 vs. 20.3 ± 11.1 *p* < 0.05; CRD11: OSA 38.8 ± 19.3 vs. 33.3 ± 14.1, *p* < 0.05) [25]. Decreasing stability of performance toward the end of the tests, assessed with end ballast (EB) measures of the CRD tests, indicated that OSA patients obtain considerably slower toward the end of tasks than control participants.

Kushida et al. demonstrated that CPAP use resulted in mild, transient improvement in the most sensitive measures of executive and frontal-lobe function for those with severe disease, which suggests the existence of a complex OSA–neurocognitive relationship. (PFN-TOTL: Baseline 23.32, after 6 months CPAP 23.48; BSRT-SR: Baseline 49.72, after 6 months CPAP 54.09) [26].

### 3.5. Patients’ Comorbidities

In this review, most articles did not include the evaluation of other comorbidities of OSA patients, particularly the analysis of their possible modification after CPAP treatment [8,11,19,23,27,28,29]. In this regard, Lim et al. defined concomitant antihypertensive drugs as exclusion criterion [18].

Interestingly, Kanbay et al. analyzed the correlation between the presence of associated comorbidities and respiratory outcomes in the two enrolled OSAS and control groups, reporting non-significant differences (hypertension *p*: 0.21; coronary heart disease *p*: 0.16; asthma *p*: 0.51; smoking status (*p* = 0. 38) However, the authors hypothesized OSAS as an independent risk factor for cognitive dysfunction, as assessed by the authors by MMSE scores and low serum IGF-1 levels [26].

This concept was later confirmed by Torelli et al., who showed, despite the higher prevalence of comorbidities (hypertension, diabetes, hypercholesterolemia, history of smoking, and drug use) in the OSA group, no significant differences with the control group (*p* > 0.05) [24].

## 4. Discussion

Continuous Airway Pressure represents an effective method of treatment of obstructive sleep apnea as it allows to obtain the maintenance of the patency of the upper airways by reducing the episodes of apnea and, therefore, sleep interruption [16,30]. It allows to obtain a better quality of sleep and consequently a reduction in daytime sleepiness.

Several studies in the literature have asserted that OSA respiratory disturbance indices are associated with the severity of cognitive dysfunction, demonstrating a positive linear correlation between AHI, ESS, ODI indices, and mnemonic functions, the extent of cerebrovascular impairment and behavior [27,28,29,30,31]. Bardwell et al. reported, in 112 patients with OSA, the severity of depression correlated with age, body mass index, and sleep parameters, probably due to nocturnal hyposaturation and sleep fragmentation [32]. Alterations of the prefrontal cortex structure between nocturnal upper airway obstruction, behavioral deficits, and cognitive performance [33,34,35,36,37].

Gozal et al. asserted that intermittent hypoxia and sleep fragmentation is seen in OSA to lead to alterations in brain structure and function, which then provides a link between sleep-disordered breathing and impairment neurocognitive domains [38]. Moreover, Wang et al. assessed cognitive function using the Montreal Cognitive Assessment (MoCA) in 28 patients with OSAHS, determining the relationship between the serum level of brain-derived neurotrophic factor (BDNF) and cognitive impairment [39]. The OSAHS group demonstrated significantly decreased serum BDNF than the control group (*t* = −10,912, *p* = 0.000) and positively correlation with the MoCA score (r = 0.544, *p* = 0.000). However, the authors did not assess the role of OSA treatment on these factors.

Werli et al. assessed 202 OSA patients effectively treated; of these, 15 patients were classified as experiencing excessive residual sleepiness (RES) that was not better accounted for by another sleep disorder [40]. After CPAP treatment, significant cognitive impairment in the RES group was demonstrated, with executive functions and depression scores being the most affected variables. Patients with RES performed more poorly on executive tests when compared to the control group but showed no changes in other functions, including learning ability, attention, or memory.

Lau et al. analyzed the role of body mass index (BMI) as a predictor for dysfunction of working memory activity, reporting no significant difference between the OSA group and the control group, although the body mass index of the OSA group was significantly higher than that of the control group, t (62) = −4.83; *p* < 0.001 [8]. These data demonstrate the close relationship between cognitive functions deficit and mild to severe forms of OSAS and how effective treatment of the pathology through CPAP can lead to an improvement of these functions through the reduction in the number of sleep apnea and, therefore of the fragmentation of sleep, which allows the reduction in daytime sleepiness typical of these patients [36,37,38,39,40,41]. As shown by Lim et al., a CPAP treatment period of more than 2 weeks is required to obtain significant results on cognitive improvement [16].

### Study’s Limitations

Although this review described various outcomes of the selected comparative studies, several limitations are present. The literature to date available lacks papers comparing the different neurocognitive functions with OSAS parameters at baseline and after different treatments administered. CPAP remains the only treatment currently, while other approaches were not adequately studied.

Another limitation found is the lack of studies with larger samples, low evidence study design, and protocol frequently not standardized.

Furthermore, the relationship between comorbidities such as hypertension, dyslipidemia, diabetes, ongoing pharmacological treatments, and the severity of cognitive impairment in patients with OSA has not been reported in most papers or fully discussed.

## 5. Conclusions

Obstructive sleep apnea syndrome is strongly associated with cerebrovascular disorders, chronic neurodegenerative and inflammatory diseases, leading to a high risk of cognitive impairment in affected patients. However, the relevant literature remains doubtful to date on the efficacy of OSA treatment on cognitive functions. Although several studies have shown that continuous positive airway pressure could improve cognitive domains such as working memory, long-term verbal memory, and short-term visuospatial memory, study designs are often weak, contain a small sample size, and have an inadequate follow-up. Furthermore, insufficient evidence is reported on the efficacy of other available therapeutic approaches such as MAD or OSA surgery. Therefore, to elucidate the role of OSA therapy on cognitive performance, additional studies are needed, with more evidence to validate the alleged efficacy.

## Figures and Tables

**Figure 1 behavsci-11-00180-f001:**
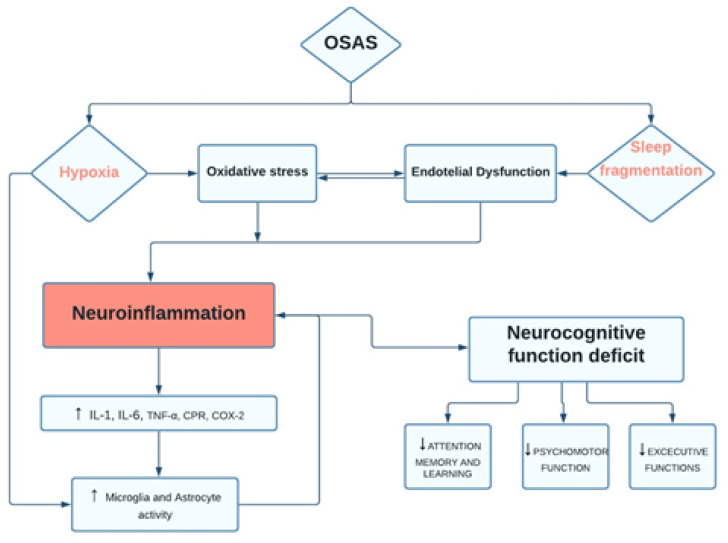
Flow-diagram. OSAS pathways and neurocognitive dysfunction.

**Figure 2 behavsci-11-00180-f002:**
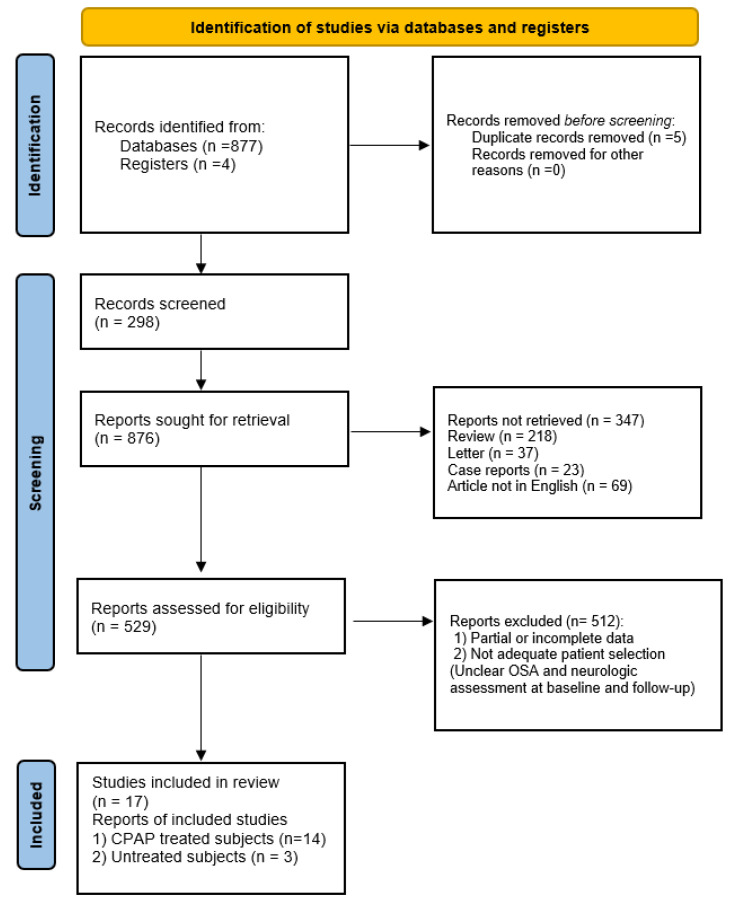
PRISMA flow diagram.

**Table 1 behavsci-11-00180-t001:** Literature data retrieved by systematic review. Abbreviations: AHI, apnea/Hypopnea index; ESS, Epworth sleepiness scale; NadirSO2, lower oxygen saturation; MeanSO2, mean oxygen saturation; RDI, respiratory disturbance index; MMSE, mini-mental state examination; WCST Wisconsin Card Sorting Test; FAS, Verbal Fluency Test; RAVL, Rey Auditory-Verbal Learning; ROCF, Rey-Osterreith Complex Figure copy; RAPM, Raven’s Advanced Progressive Matrices; SVFT, Semantic verbal fluency task; PVFT, Phonological verbal fluency task; WAIS-R, Wechsler Adult Intelligence Scale-Revised; WCST, Wisconsin Card Sorting Test; EXIT, Executive interview; PHQ-9, Patient Health Questionnaire; TMB Trail-Making Test B; MoCA Montreal cognitive assessment; PFN-TOLT, Pathfinder Number Test-Total Time; BSRT-SR, Buschke Selective Reminding Test-Sum Recall; SWMT-OMD, Sustained Working Memory Test-Overall Mid-Day Index; ACE-R Addenbrooke’s Cognitive Examination-Revised; LM, Logical Memory; TMA Trail-Making Test A; CRD, Complex Reactiometer Drenovac test battery 311 (speed of perception to visual stimulus), 411 (speed of complex psycho-motor limbs coordination), 11 (speed of solving simple arithmetic operations), * healthy group, ** OSA patient adequately treated with CPAP.

Authors	Study Design	Sample	Control Group	Age	Gender	Treatment	Sleep Parameters Pre vs. Post Treatment (mean ± SD)	Comorbidities	Questionnaire	Outcomes Pre	Outcomes Post	*p* -Value	Follow Up
Ng, S.S. et al., 2015	Prospective controlled	30 OSA	-	73.9 ± 7.5	109 M, 125 F	CPAP	AHI: 16.8 ± 14.2 Nadir: 79 ± 14	Hypertension, diabete mellitus, cardiovascular disorders	Digit span	15.2 ± 3.08	15.4 ± 4.02	*p* = 0.285	12 months
									Digit Symbol	30.4 ± 12.2	35.7 ± 15.1	*p* < 0.001	
									Stroop colour	60.2 ± 17.7	65 ± 17.6	*p* = 0.001	
Kanbay, A. et al., 2017	Prospective controlled	33 OSA	17 *	51 ± 9 vs. 47 ± 6	26 M 25 F	CPAP	AHI: 45.3 ± 30.9	Hypertension, cardiovascular disorders, asthma	MMSE	23.5 ± 3.6 vs. 28.1 ± 1.4	28.1 ± 1.6 vs. 28.1 ± 1.6	*p* = 0.001	3 months
Werli, K.S. et al., 2016	Prospective controlled	15 OSA RES	15 **	51.0 ± 8.4 vs. 51.8 ± 8.2	19 M 11 F	CPAP	ESS: 15 ± 2.5 AHI: 56.18 ± 27.55 meanSO2: 93.58 ± 2.93	-	WCST categories	-	1.6 ± 1.4 vs. 3 ± 1.4	*p* = 0.04	12 months
									FAS	-	25.5 ± 5.3 vs. 30.7 ± 7.32	*p* = 0.04	
Torelli, F. et al., 2011	Prospective controlled	16 OSA	14 *	55.8 ± 6.7 vs. 57.6 ± 5.1	22 M 8 F	untreated	AHI: 52.2 ± 2.6 meanSO2: 92.0 ± 3.1	Hypertension, Diabetes, Hypercholesterolemia, Smoking	MMSE	29.5 ± 0.8 vs. 29.6 ± 0.6	--------------	*p* = 0.60	12 months
									RAVL	40.9 ± 5.4 vs. 45.9 ± 6.4	-	*p* = 0.026	
									Digit span	5.6 ± 0.6 vs. 5.9 ± 0.4	-	*p* = 0.23	
									Visual memory	20.4 ± 1.2 vs. 19.7 ± 1.7	-	*p* = 0.22	
									Copy drawings	9.9 ± 2.1 vs. 10.1 ± 1.2	-	*p* = 0.72	
									ROCF	30.4 ± 6.1 vs. 33.9 ± 3.0	-	*p* = 0.06	
									RAPM	29.7 ± 3.9 vs. 31.7 ± 2.6	-	*p* = 0.11	
									SVFT	39.7 ± 0.5 vs. 39.9 ± 0.3	-	*p* = 0.11	
									PVFT	26.7 ± 8.8 vs. 28.5 ± 6.6	-	*p* = 0.56	
									Stroop Test	40.3 ± 13.1 vs. 33.9 ± 5.0	-	*p* = 0.10	
Lau, E.Y. et al., 2010	Prospective controlled	37 OSA	27 *	57.9 ± 9.5 vs. 56.7c10.5	22 M 15 F	CPAP	ESS: 14.4 ± 5.2 vs. 8.3 ± 4.5 RDI: 42.2 ± 2.9 vs. 1.7 ± 1.5 meanSO2: 93.7 ± 3.5 vs. 95.7 ± 1.6	-	WAIS-R Vocabulary	-	56.6 ± 9.4 vs. 61.3 ± 5.9	*p* = 0.017	3 months
									WAIS-R Block Design	-	30.8 ± 8.3 vs. 33.8 ± 10.1	*p* = 0.193	
									Digit Span	-	16.1 ± 4.0 vs. 16.9 ± 0.2	*p* = 0.474	
									Stroop Color-Word	-	39.0 ± 7.7 vs. 42.5 ± 7.8	*p* = 0.082	
									WCTS	-	4.8 ± 1.6 vs. 5.6 ± 1.1	*p* = 0.028	
									Rey-O Recall	-	17.3 ± 6.1 vs. 19.2 ± 6.4	*p* = 0.229	
Akmal, M.K. et al., 2013	Cross-sectional study	20 OSA	-	43.6 ± 4.12	14 M vs. 6 F	CPAP	-	Psychiatric comorbidities	EXIT25	24.5 ± 5.82	39.8 ± 5.41	*p* < 0.001	1 months
Edwards, C. et al., 2015	Cross-sectional study	228 OSA	-	52 ± 15.57	243 M vs. 183 F	CPAP	AHI: 46.7 ± 27.4 vs. 6.5 ± 1.6	-	PHQ-9 Depression Scale	11.3 ± 6.1	3.7 ± 2.9	*p* < 0.001	3 months
Barnes et al., 2004	Randomised controlled trial	114 OSA	-	47 ± 0.9	91 M vs. 23 F	CPAP vs. MAS or Placebo	AHI: 22.2 ± 1.5 ESS: 10.2 ± 0.5	Hypertension	Stroop color association test	-	9.3 ± 0.9 vs. 10.3 ± 0.9 vs. 9.2 ± 0.9	*p* < 0.001	3 months
									Digit span backward	-	4.6 ± 0.1 vs. 4.6 ± 0.1 vs. 4.8 ± 0.1	*p* < 0.05	
									Digit symbol substitution task	-	47.3 ± 0.4 vs. 47.5 ± 0.4 vs. 46.8 ± 0.4	*p* < 0.05	
Castronovo et al., 2009	Prospective controlled	17 OSA	15 *	43.93 ± 7.8	32 M	CPA	AHI: 61.35 ± 97.7 vs. 9.8 ± 1.6 NadirSO2: 72.45 ± 7.69 vs. 91.2 ± 1.3 ESS: 12.0 ± 5.18 vs. 3.08 ± 2.24	-	Rey’s List (learning)	58.0 ± 7.1 vs. 48.54 ± 10.15	57.54 ± 8.36 vs. 48.54 ± 10.15	*p* < 0.001	3 months
									Corsi	5.23 ± 1.09 vs. 6.53 ± 0.91	6.31 ± 0.85 vs. 6.53 ± 0.91	*p* = 0.002	
									TMB	82.15 ± 26.16 vs. 59.4 ± 14.16	78.85 ± 23.42 vs. 59.4 ± 14.16	*p* < 0.001	
									Stroop	5.08 ± 3.32 vs. 0.73 ± 1.03	0.83 ± 1.53 vs. 0.73 ± 1.03	*p* < 0.001	
Wang, W.H. et al., 2012	Randomised controlled trial	28 OSA	14 *	44.93 ± 2.98	42 M	untreated	AHI: 49.63 ± 28.56	-	MoCA	24.04 ± 1.75 vs. 28.57 ± 1.09	-	*p* < 0.01	
Kushida, C.A. et al., 2012	Randomised controlled trial	1098 OSA	-	52.2 ± 12.2	719 M 379 F	442 activeCPAP vs. 401sham CPAP	AHI: 30.7 ± 24.9 NadirSO2: 81 ± 7.6	-	PFN-TOTL	23.32 vs. 23.08	23.48 vs. 23.01	*p* = 0.2103	6 months
									BSRT-SR	49.72 vs. 48.86	54.09 vs. 54.28	*p* = 0.7569	
									SWMT-OMD	0.035 vs. −0.074	0.072 vs. 0.018	*p* = 0.2254	
Rosenzweig, I. et al., 2016	Randomised controlled trial	68 OSA	35 *	47.6 ± 11.1	80 M 23 F	CPAP	AHI: 36.58 ± 27.15	-	ACE-R	90.55 ± 1.11 vs. 94.91 ± 0.99	91.86 ± 2.44 vs. 90.70 ± 1.85	*p* = 0.220	1months
									Immediate LM	36.36 ± 1.32 vs. 47.06 ± 1.84	44.43 ± 1.99 vs. 41.70 ± 2.28	*p* = 0.338	
									TMB	62.02 ± 3.65 vs. 41.23 ± 2.0	51.05 ± 3.68 vs. 61.53 ± 4.81	*p* = 0.0017	
									TMA	27.34 ± 1.04 vs. 24.12 ± 1.18	24.26 ± 1.28 vs. 28.02 ± 1.56	*p* = 0.937	
Lim, W. et al., 2007	Randomised controlled trial	46 OSA	-	46.7 ± 2.4	-	CPAP-Oxygen-PlaceboCPAP	AHI: 63.5 ± 7.8 ESS: 11.2 ± 1.0 MeanSO2: 93.1 ± 1.1	Hypertension	Letter/Number Sequencing	11.0 vs. 11.1 vs. 11.7	11.9 vs. 11.7 vs. 12.9	*p* = 0.005	2 weeks
									Digit Span Total	18.6 vs. 19.3 vs. 21.2	26.4 vs. 21.3 vs. 22.5	*p* = 0.091	
									Digit Vigilance	5.6 vs. 8.9 vs. 14.1	7.2 vs. 7.3 vs. 10.6	*p* = 0.196	
									Stroop Color-Word	37.7 vs. 40.1 vs. 37.9	37.3 vs. 44.2 vs. 41.9	*p* = 0.007	
Turner et al., 2019	Prospective cohort study	16OSA	-	36–80	15 M 1 F	CPAP	ESS: 9.31 ± 5.87 vs. 5.69 ± 3.44	Epilepsy	Digit Span Forwad	5.62 ± 1.02	5.81 ± 1.05	*p* = 0.61	3 months
									Digit Span Backward	4.13 ± 0.7	05.12 ± 1.02	*p* = 0.004	
									ROCF	19.08 ± 7.32	21.09 ± 7.69	*p* = 0.48	
									Corsi Span	4.94 ± 0.9	5.6 ± 0.6	*p* = 0.02	
									Short story test	10.19 ± 3.72	13.84 ± 2.89	*p* = 0.004	
									Attentional Matrices	54.0 ± 6.0	55.06 ± 3.85	*p* = 0.55	
Canessa, N. et al., 2010	Prospective controlled	17 OSA	15 *	44 ± 7.63	32 M	CPAP	AHI: 55.83 ± 19.08 vs. 2.5 ± 2.4 MeanSO2: 70.41 ± 9.13 vs. 91.4 ± 1.9	Hypertension	MMSE	29.35 ± 1.05 vs. 30.00	29.75 ± 0.57	Ns	3 months
									Raven	31.70 ± 3.90 vs. 34.6 ± 1.29	33.25 ± 2.46	*p* = 0.03	
									Digit Span forward	5.58 ± 1.00 vs. 6.93 ± 0.70	6.56 ± 0.81	*p* = 084	
									Rey-list recall	48.70 ± 9.67 vs. 13 ± 1.96	58.18 ± 7.92	*p* < 0.001	
Liguori, C. et al., 2017	Prospective controlled	25 OSA vs. 10 OSA-CPAP	15 *	67.96 ± 7.92	26 F 14 M	CPAP	AHI: 36.34 ± 11.42 vs. 3.14 ± 1.54 MeanSO2: 92.37 ± 1.97 vs. 95 ± 0.82	Hypertension	I-RAVL	-	42.58 ± 2.50 vs. 46.7 ± 1.49 vs. 49.07 ± 3.22	*p* < 0.001	12 months
									Raven	-	26.73 ± 69.33 vs. 33 ± 1.41 vs. 33.07 ± 0.80	*p* < 0.0001	
									Stroop color/word test	-	33.96 ± 4.15 vs. 29.2 ± 1.5 vs. 26.57 ± 2.22	*p* < 0.0001	
Lusic Kalcina, L. et al., 2020	Prospective controlled	103 OSA	103 *	57.14 ± 11.31	206 M	untreated	ESS: 8.65 ± 4.5 AHI: 45.02 ± 14.09	Hypertension, diabetes, cardiovascular disorders, depression, arthritis, thyroid disease	CRD11-EB	38.8 ± 19.3 vs. 33.3 ± 14.1	-	*p* = 0.028	
									CRD311-EB	5.2 ± 1.8 vs. 4.5 ± 1.3	-	*p* = 0.003	
									CRD411-EB	25.1 ± 17.9 vs. 20.3 ± 11.1	-	*p* = 0.038	

**Table 2 behavsci-11-00180-t002:** Description of neurocognitive tests.

Questionnaire	Features
Digit Span	Measures cognitive attention abilities, working memory (central executive), and inhibition. Participants are presented with a random series of digits and are asked to repeat them in either the order presented (forward span) or in reverse order (backward span)
Digit Symbol	The test assesses brain damage, dementia, and depression, consisting of digit-symbol pairs followed by a list of digits.
Stroop tests	The test is used to examine the effects of interference on reading ability. Contains three parts: word page, color page, and word-color page, each with five columns containing 20 items. The subject’s task is to look at each sheet and move down the columns, reading words or naming the ink colors as quickly as possible, within a given time limit (45 s).
MMSE	The test included 11 questions in five categories as follows: orientation, registration memory, attention, and calculation, recall memory, and language.
WCSTcategories	This test evaluates the following functions: formation of concepts and problem solving, mental flexibility, abstraction-reasoning, and strategizing.
FAS	Test that evaluates the capacity of evoking words (under delimited conditions) and problem-solving strategies. The outcome variable is the number of words remembered.
RAVL	It is a list of 15 words read to the subject five times. Measures immediate memory, learning efficiency, interference effects, and recall after short and long periods.
Visual memory	Subjects are required to view a simple figure for 3 s and then recognize it in a multiple-choice condition to evaluate short memory.
Copy drawing	This task requires reproducing a geometrical figure both by freehand and by joining landmarks already traced on the sheet to evaluate construtional praxia.
ROCF	In this test, the subject is asked to reproduce a bidimensional complex figure from memory without forewarning, 15 min after copy, to evaluate short and long memory
RAPM	It is a set of 3 subtests (labeled A, Ab, and B) to evaluate non-verbal intelligence, visual processing speed, cognitive speed, and flexibility. It consists in choosing from a set of distractors the item logically missing in a given visual/spatial set.
SVFT	Subjects have to produce as many words as they can that fall into three semantic categories, in a time limit of 1 min per sub-test, to evaluate lenguage.
PVFT	Subjects have to produce as many words as they can, beginning with a given letter (A, F, S), in a time limit of 1 min per sub-test, to evaluate executive function.
WAIS-R vocabulary	Twelve vocabulary words were presented, and participants were asked to define this word.
WAIS-R block design	Determine the clinical value of measuring visuospatial abilities. The patient use hand movements to rearrange blocks with various color patterns on different sides to match a pattern.
Stroop color-word	The test measures selective and focused attention, cognitive flexibility, and inhibition.
WCTS	Test to examine the patient’s frontal functions; used to evaluate flexibility in the choice of problem-solving strategies and used to evaluate the inability to abstraction as well as perseveration.
Rey-O Recall	Examinees are asked to reproduce a complicated line drawing, first by copying it freehand (recognition) and then drawing from memory (recall).
EXIT 25	The test consists of 25 items to assess executive functions in people with normal cognition or impairment to identify specific subtypes of mild cognitive impairment and the risk of dementia conversion.
PHQ9 depression scale	It is an instrument for making criteria-based diagnoses of depression. Higher PHQ-9 scores are associated with decreased functional status and increased symptom-related difficulties, sick days, and healthcare utilization.
Corsi	Test to assess visuospatial short-term memory.
TMB	Assesses executive abilities, including setting-shifting and mental flexibility.
MoCA	The test evaluated executive function, naming, attention, calculation, language, abstraction, memory, and orientation.
PFN-TOLT	The test assesses attention and psychomotor function and comprises the total time for the participant to scan, locate, and connect numbers in sequence.
BSRT-SR	The test assesses verbal learning and memory and consists of the total words recalled across six selective reminding trials.
SWMT-OMD	The test assesses an executive and frontal-lobe function component by requiring the participant to compare the spatial position of a stimulus with its position on a previous trial (*n*-backtest), pressing one button if the spatial position was the same as that on the previous trial or a second button if it differed.
ACE-R	Provides evaluation of six cognitive domains: orientation, attention, memory, verbal fluency, language, and visuospatial ability. It is useful for detecting dementia and mild cognitive impairment score.
Immediate LM	This test assessed the patient’s ability to remember two short stories presented orally, and it is a measure of verbal memory.
TMA	The test measures visual attention and processing speed.
Letter/number sequencing	It is a test that measures an individual’s short-term memory skills in processing and re-sequence information.
Digit Vigilance	It measures vigilance during rapid visual tracking and accurate selection of target stimuli. It focuses on alertness and vigilance while placing minimal demands on two other components of attention: selectivity and capacity.
Corsi span	It is a test to assess a visuospatial memory.
Short story test	It assesses long-term verbal memory: a short story is read to the subject with the instruction to repeat, immediately afterward, everything they remember; then, the story is read again. After 10 min, the patient is asked to repeat the story once again.
Attentional matrices	Test to assess attention involves the use of rows of numbers randomly interspersed with a designated target number or numbers; the patient is instructed to cross out all target numbers in three matrices («5» in I, «2–6» in II, «1–4–9» in III), arranged in a random sequence, within a time limit of 45 s.
Raven	It is a non-verbal test used to measure one’s ability to use reasoning and logical ability.
CRD TEST	The chronometric instrument can measure: speed of solving simple arithmetic operations (CDR11); the speed of perception to a visual stimulus (CRD311); the speed of complex psycho-motor limbs coordination (CRD411).

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
