# Peer review of "Neurocognitive Performance Improvement after Obstructive Sleep Apnea Treatment: State of the Art"

_behavsci, 2021, doi:10.3390/bs11120180_

Round 1

Reviewer 1 Report

Behavioral Sciences

Full Title:

Neurocognitive Performance improvement after obstructive sleep apnea treatment: state of the art

Article Type

Review

Revision comments

General overview

The authors proposed to review the last 20 years’ literature on the effects of OSA treatment on neurocognitive performance. Overall, the paper is well structured, easy to read, and provides a good overview of the state of the art and missing gaps/future directions. Yet, there are some aspects that should be improved, namely:

Abstract

  • “Obstructive Sleep Apnea (OSA) syndrome is a respiratory sleep disorder characterized by partial or complete episodes of upper airway collapse with reduction or complete cessation of airflow, affecting the neurocognitive performance with several mechanisms such as intermittent hypoxemia, sleep deprivation, hypercapnia, disruption of the hypothalamic-pituitary-adrenal axis.”

I recommend being more cautious. The relationship between OSA and neurocognitive dysfunction is still controversial as also observed in this review.

Introduction

  • Figure 1: Sleep fragmentation should be also added in the flow-diagram as an important hallmark of OSA potentially contributing to neurocognitive function deficits. Eventually replacing oxidative stress with an arrow towards neurocognitive function deficit, leaving endothelial dysfunction on the right side.

Material and Methods

  • Inclusion criteria includes “The studies reported detailed information on pre and post-treatment OSAS outcomes, validated questionnaires on neurocognitive performance at baseline and after treatment, and patient’s comorbidities”. Why did the authors ended up including three studies assessing neurocognitive deficits only in untreated patients then? And if patient’s comorbidities should be reported, why did the authors not commented about it at any point throughout the manuscript? Do the patients of the selected studies have any comorbidities? Do these change with treatment as well? Can these be cofounding factors? Presence of comorbidities should be discriminated, at baseline and after treatment (whenever possible), and its impact should be included in the discussion section.

Results

  • Figure 2 (PRISMA): “2) Not adequate patient selection”. What does this mean? It is not clear, the criteria used should be described.

  • Table 1:

  1. The table should be more concise. E.g. sample column, gender (n vs %), treatment, follow-up (1 month = 4 weeks, 1 term should be chosen)
  2. It would be important to include a control group column. One cannot really understand what is the control group in some cases. E.g. In studies that did not evaluated patients after treatment, is the control group healthy subjects? How many? Include n.
  3. In sample column, separate groups by row to make it clearer. For example, in Kanbay A et al., 2017: 314 OSA (33 treated vs 17 CG), what happens to the remaining patients? Were these not treated? Is should be clear. And what is CG? It is not defined in the table legend.
  4. RES is also not defined in the table legend.
  5. It would be important to know the clinical characteristics of each study cohort: BMI; comorbidities; medication; OSA severity (very important and not really commented!), evaluated OSA/sleep parameters (AHI, arousals, mean/min O2 levels; desaturations), sleepiness questionnaires, at baseline and after treatment (whenever possible). Cohorts characteristics could be in a separate table, including age and gender.
  6. Make sure that the legend includes all abbreviations present in the table.

  • The questionnaires used in the different studies should be better explained for the reader to understand better the meaning of the presented values. An additional table could be created with the different tests, brief explanation and utility (used to assess executive function? Attention? Associate to each results section) and scoring.

  • Describe also the severity and characteristics of the patients included in each results section.

Discussion

  • Line 213-214: “The OSAHS group had significantly decreased total MoCA compared to control group.” Suggesting… ?
  • Line 219-220: “In this study, the OSA group was not treated and shows how Sleep Apnea Disorder causes cognitive deterioration if not adequately treated”. This sentence should be reformulated. It does not show “how” and it is an extrapolative conclusion.  Suggesting that OSA may contribute to cognitive deterioration if not adequately treated.
  • The relationship between OSA severity/evaluated clinical parameters and cognitive deficits should be discussed.
  • Comment on the impact of cofounding factors such as comorbidities (some have been reported to ameliorate with OSA treatment, so it may also contribute to ameliorations at the neurological level), medication (patients before treatment may use sleep pills more often than during treatment which impacts on neurocognitive function), BMI (some patients loose weight upon OSA treatment), age (some studies have older cohorts or include a wider range of ages -> efficacy of treatment may be affected by age), gender differences, disease history (patients with untreated OSA for longer periods may show more cognitive deficits that patients that were treated faster) and heterogeneity (different symptoms and clinical manifestations). Contribute to the variability of results and conclusions observed in the different studies.

Conclusion

Reinforce the importance of better understanding the impact of OSA and its treatment on cognitive impairment and the need of more studies to obtain conclusions.

Reviewer 2 Report

The manuscript deals with an important topic that for more than two decades has aroused interest in Europe, sleep apnea was previously a disease that was only treated for those of purchasing power, today it is treated throughout the population for its prevalence and association with unhealthy lifestyles,   this part is not worked on in the introduction, just as it does not refer to authors referring to it such as Durán J, etc ...

The introduction is not done in depth and should be improved, taking into account the pioneering treatments that currently exist, nor does it refer to the school of CPAP patients.

Regarding the methodology, it seems correct to me, it uses the PICO strategy, in addition to making a flowchart explicit.

It should delve into the previous phase, how and why it develops those inclusion criteria where it is limited in different choices:

Why only evidence in the English language??? It's a bias

Why not qualitative studies? Bias

It is a big mistake that the authors describe in the manuscript the reference to researchgate, which is a good application for contact between researchers, but not with the character of this journal.

You must review the biliography, it has some faults

Round 2

Reviewer 1 Report

Prior publication, I would recommend authors to check spelling mistakes (For example: sleep fragmentation in figure 1, not adequate patient selection in figure 2) and english language with native speaker. 

Author Response

Dear reviewer,

thanks for the revisions. We've corrected all the grammatical errors and performed English editing. We've attached the clear and highlighted copy with the modified sentences.

Best regards.
